# Conformational changes in the Ebola virus membrane fusion machine induced by pH, Ca²⁺, and receptor binding

**Dibyendu Kumar Das**[1,2]*, **Uriel Bulow**[1], **William E. Diehl**[3], **Natasha D. Durham**[1,4], **Fernando Senjobe**[1], **Kartik Chandran**[5], **Jeremy Luban**[3], **James B. Munro**[1,4,6]*

**1** Department of Molecular Biology and Microbiology, Tufts University School of Medicine and Sackler School of Graduate Biomedical Sciences, Boston, Massachusetts, United States of America, **2** Department of Biological Sciences and Bioengineering, Indian Institute of Technology, Kanpur, India, **3** Program in Molecular Medicine, University of Massachusetts Medical School, Worcester, Massachusetts, United States of America, **4** Department of Microbiology and Physiological Systems, University of Massachusetts Medical School, Worcester, Massachusetts, United States of America, **5** Department of Microbiology and Immunology, Albert Einstein College of Medicine, Bronx, New York, United States of America, **6** Department of Biochemistry and Molecular Pharmacology, University of Massachusetts Medical School, Worcester, Massachusetts, United States of America

* james.munro@umassmed.edu (JBM); dkdas@iitk.ac.in (DKD)

**Data Availability Statement:** All relevant data are within the paper and its Supporting Information files.

## Abstract

The Ebola virus (EBOV) envelope glycoprotein (GP) is a membrane fusion machine required for virus entry into cells. Following endocytosis of EBOV, the GP1 domain is cleaved by cellular cathepsins in acidic endosomes, removing the glycan cap and exposing a binding site for the Niemann-Pick C1 (NPC1) receptor. NPC1 binding to cleaved GP1 is required for entry. How this interaction translates to GP2 domain-mediated fusion of viral and endosomal membranes is not known. Here, using a bulk fluorescence dequenching assay and single-molecule Förster resonance energy transfer (smFRET)-imaging, we found that acidic pH, Ca²⁺, and NPC1 binding synergistically induce conformational changes in GP2 and permit virus-liposome lipid mixing. Acidic pH and Ca²⁺ shifted the GP2 conformational equilibrium in favor of an intermediate state primed for NPC1 binding. Glycan cap cleavage on GP1 enabled GP2 to transition from a reversible intermediate to an irreversible conformation, suggestive of the postfusion 6-helix bundle; NPC1 binding further promoted transition to the irreversible conformation. Thus, the glycan cap of GP1 may allosterically protect against inactivation of EBOV by premature triggering of GP2.

## Introduction

Recent and ongoing outbreaks of Ebola virus (EBOV) disease in western and central Africa have resulted in unprecedented loss of life. The persistent threat of EBOV infection underscores the importance of establishing new strategies for therapeutic and preventative measures. Achieving this goal will benefit from a better understanding of the mechanism by which EBOV enters cells.

**Funding:** This work was supported by National Institutes of Health grants DP2AI124384 (to J.B.M) and R01AI148784 (to J.L.). The funders had no role in study design, data collection and analysis, decision to publish, or preparation of the manuscript.

**Competing interests:** The authors have declared that no competing interests exist.

**Abbreviations:** Blam, β-lactamase; EBOV, Ebola virus; eRF1, eukaryotic release factor 1; FL, fusion loop; FRET, Förster resonance energy transfer; GP, EBOV envelope glycoprotein; GPΔmuc, GP with the mucin-like domain deleted; HA, hemagglutinin; HIV, human immunodeficiency virus; HMM, hidden Markov modeling; MD, molecular dynamics; NAADP, nicotinic acid adenine dinucleotide phosphate; NPC1, Niemann-Pick C1; PEG, polyethylene glycol; RuV, Rubella virus; SERM, selective estrogen receptor modulator; smFRET, single-molecule Förster resonance energy transfer; sNPC1-C, soluble domain C of NPC1; TCO*, *trans*-cyclooct-2-ene-L-lysine; TIRF, total internal reflection fluorescence; TPC, two-pore channel; VSV, vesicular stomatitis virus; 6HB, 6-helix bundle.

The EBOV envelope glycoprotein (GP) resides on the surface of the virion and mediates entry into host cells [1,2]. In the virus-producing cell, the GP0 polypeptide is posttranslationally cleaved by the host furin protease into disulfide-linked GP1 and GP2 domains [3]. GP1 facilitates attachment to the host cell. GP2 promotes fusion of the viral and endosomal membranes by a poorly defined mechanism. The EBOV virion is internalized by macropinocytosis and trafficked to the late endosome [4–6]. The acidic environment activates the cellular cathepsin B and L proteases, which cleave the mucin-like and glycan cap domains from GP1 [7,8], exposing the underlying binding site for the Niemann-Pick C1 (NPC1) receptor [9–12]. Whether the acidic environment of the late endosome is important in triggering fusion mediated by intact trimeric GP beyond activation of the cathepsins is unknown. However, studies of the isolated fusion loop (FL) [13], which resides near the N terminus of GP2 in the intact trimer, and thermodynamic measurements of the GP ectodomain have implicated a role for acidic pH [14]. NPC1 is essential for EBOV entry [15,16], but the underlying role of GP-NPC1 interaction in promoting the activation of GP2 for membrane fusion is unknown.

In addition to NPC1, EBOV entry into host cells also requires the activity of endosomal two-pore $Ca^{2+}$ (TPC) channels [17], which are triggered by nicotinic acid adenine dinucleotide phosphate (NAADP) to release $Ca^{2+}$ from endosomes and lysosomes [14]. Inhibiting TPC function prevented EBOV infection, which implicates endosomal $Ca^{2+}$ in EBOV entry. Several approved drugs, including selective estrogen receptor modulators (SERMs) possess anti-EBOV activity [18]. SERMs caused endo-lysosomal $Ca^{2+}$ accumulation, further suggesting a role for $Ca^{2+}$ in EBOV entry [19]. The role of $Ca^{2+}$ in EBOV entry remains unknown, and the nature of its interaction with the intact GP trimer has not been elucidated. However, studies of the isolated FL of GP suggest that $Ca^{2+}$ may facilitate interaction with the target membrane [20].

Crystallographic structural data depict GP in a prefusion conformation in which the FL in GP2 resides in a hydrophobic cleft within the neighboring protomer [17–20]. Structures of GP in the postfusion conformation indicate a refolding of GP2, resulting in formation of a 6-helix bundle (6HB) [21,22]. Thus, the endpoints of the GP-mediated membrane fusion reaction have been described in atomic detail. But the sequence of events connecting these states and the presence of any functional intermediate states has not been described, although this mechanism may resemble that of influenza hemagglutinin (HA), the canonical class-I viral fusogen [23,24]. According to the model of HA-mediated membrane fusion, the FL must be released from the hydrophobic cleft and extend away from the surface of the virion, gaining access to the target endosomal membrane. The FL can then insert into the endosomal membrane, forming an extended intermediate. This putative intermediate then collapses, drawing the viral and endosomal membranes together, promoting their fusion. Thus, large-scale refolding of GP is postulated during viral fusion, which includes dramatic movements of the FL. Until recently, methodology that permits direct visualization of conformational changes in GP related to fusion has not been available.

Here, we probed the role of $Ca^{2+}$, acidic pH, and NPC1 binding in the mechanism of GP-mediated membrane fusion. We first developed a bulk fluorescence-based virus-liposome lipid mixing assay. This approach demonstrated that NPC1 is essential to promote even a low level of lipid mixing, with acidic pH and $Ca^{2+}$ increasing both the extent and kinetics of GP-mediated lipid mixing. We then developed an smFRET imaging assay to directly visualize the conformational changes in the GP2 domain, with a focus on visualizing the motion of the N terminus and the proximal FL. Our smFRET results showed that acidic pH and $Ca^{2+}$ synergistically promote conformational changes in GP2, which involve the movement of the FL and the N terminus of GP2 away from the surface of the virion and toward the target membrane. These conformational changes remain fully reversible until the glycan cap is removed from

GP1, at which point GP can transition to an irreversible conformation, which may be the postfusion 6HB. NPC1 binding accelerates the transition to the irreversible conformation. These data suggest a fusion model in which the low pH and $Ca^{2+}$ concentration of the late endosome are critical to ensuring efficient EBOV entry. Furthermore, in addition to occluding the receptor-binding site, the glycan cap allosterically regulates the conformational dynamics of GP2. The necessity of glycan cap removal and subsequent NPC1 binding for efficient membrane fusion may prevent premature triggering of GP to the inactive postfusion conformation in the presence of $Ca^{2+}$ at acidic pH.

## Results

### Low pH and $Ca^{2+}$ accelerate GP-mediated lipid mixing

We first reconstituted virus-liposome lipid mixing using a fluorescence-based assay, an approach originally established for studying influenza fusion [21]. We formed pseudovirions using the human immunodeficiency virus (HIV) core and EBOV GP from the Mayinga strain. We genetically removed the mucin-like domain from GP (GPΔmuc), which is dispensable for EBOV entry [22]. The glycan cap was then proteolytically removed, forming $GP^{CL}$, which is competent for binding NPC1 (Materials and methods) [9]. The viral membrane was labeled with a self-quenching concentration of the lipophilic fluorophore DiO. Membrane-labeled virions maintained approximately 75% infectivity compared with unlabeled virions (S1A Fig). The labeled virions were introduced to liposomes coated with the soluble domain C of NPC1 (sNPC1-C). At neutral pH, we observed slow DiO dequenching, indicating that $GP^{CL}$ is capable of inefficiently mediating lipid mixing under these conditions (Fig 1A). Inclusion of a physiological concentration of $Ca^{2+}$ (0.5 mM [23]) increased the rate of dequenching at neutral pH, although the extent of dequenching remained limited (Fig 1A). In agreement with live-cell imaging data [15], no dequenching was seen in the absence of sNPC1-C (Fig 1A) or prior to proteolytic removal of the glycan cap (S2A Fig). This confirms that the observed dequenching arises because of lipid mixing mediated by GP. Decreasing the pH to 6.2 and 5.2 resulted in increases in the extent of dequenching. Even at pH 5.2, no dequenching occurred in the

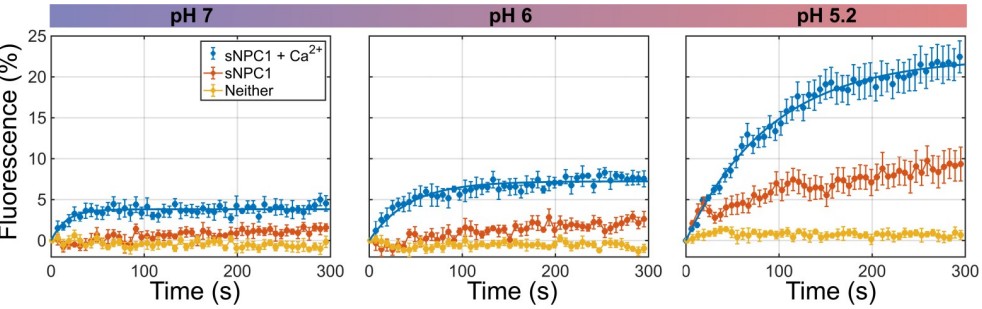

**Fig 1. Robust EBOV GP-mediated virus-liposome lipid mixing requires acidic pH, $Ca^{2+}$, and NPC1.** Fluorescence dequenching reports on lipid mixing between labeled $GP^{CL}$-containing pseudovirions and unlabeled liposomes. Fluorescence dequenching data acquired at the indicated pH are presented as the percentage of dequenching seen upon addition of 1% triton X-100, which solubilizes the viral membrane leading to maximal dequenching. Data are shown for liposomes without (yellow) or with (orange) sNPC1-C and in the additional presence of 0.5 mM $CaCl_2$ (blue). The data are displayed as the average of 4 independent measurements, with error bars reflecting the standard deviation. All dequenching data acquired in the presence of sNPC1-C and $Ca^{2+}$ are fit to the exponential function $A(1-e^{-kt})$ with fitting parameters at pH7: $A = 0.038 \pm 0.001$, $k = 0.06 \pm 0.02$; pH6: $A = 0.072 \pm 0.002$, $k = 0.023 \pm 0.003$; and pH5.2: $A = 0.224 \pm 0.006$, $k = 0.011 \pm 0.001$. The error bars in the fits represent 95% confidence intervals. All fits had $R^2 > 0.9$. Fits were not well determined for the data lacking $Ca^{2+}$ or sNPC1-C because of the limited observation window. Numeric lipid mixing data are provided in S1 Data. EBOV, Ebola virus; GP, EBOV envelope glycoprotein; NPC1, Niemann-Pick C1; sNPC1-C, soluble domain C of NPC1.

absence of sNPC1-C. The rate of dequenching remained equivalently slow in the absence of $Ca^{2+}$, irrespective of pH (Fig 1). The presence of equivalent concentrations of other divalent cations, $Mg^{2+}$ or $Zn^{2+}$, failed to accelerate dequenching at pH 5.2 in the presence of sNPC1-C (S2C Fig). Comparably robust dequenching was seen over a range of physiological $Ca^{2+}$ concentrations (0.1–0.5 mM) at pH 5.2 in the presence of sNPC1-C (S2B Fig). However, the extent of dequenching was precipitously reduced in the presence of elevated $Ca^{2+}$ concentrations ranging from 1 to 10 mM (S2B Fig). Consistently, $Ca^{2+}$-channel antagonists and SERMs, which cause an accumulation of endo-lysosomal $Ca^{2+}$, inhibited GP-mediated virus entry (S3 Fig) [17–19]. These data demonstrate that NPC1 binding is essential for even a low level of GP-mediated lipid mixing and that physiological $Ca^{2+}$ concentrations and acidic pH increase the extent and kinetics of lipid mixing.

## Site-specific attachment of fluorophores to GP

We next sought to understand the individual and combined effects of pH, $Ca^{2+}$, and NPC1 binding on GP conformational changes related to membrane fusion. To this end, we developed an smFRET imaging assay similar to that developed for visualizing influenza HA conformational changes [24] (Fig 2A). In the context of GPΔmuc, we attached 1 fluorophore at position 501 at the N terminus of GP2, proximal to the FL [25,26]. We attached the second fluorophore at position 610, between helices α4 and α5 and adjacent to the $CX_6CC$ motif in GP2 (Fig 2B) [27]. We chose these sites to track the predicted conformational changes in GP2 during membrane fusion that lead to displacement of the N terminus and the FL with respect to the base of GP2. Structural data suggest a significant increase in the distance (approximately 60–70 Å) between the fluorophores after transition from the prefusion conformation to the postfusion 6HB conformation, which we anticipated would yield a decrease in FRET efficiency [25,27–30]. To facilitate fluorophore attachment, we incorporated 2 trans-cyclooct-2-ene-L-lysine (TCO*) amino acids at positions 501 and 610 through amber stop codon suppression [31] (GP*; S4 and S5 Figs). We then formed pseudovirions with the HIV core and either GP or GP*. The pseudovirions were labeled with Cy3- and Cy5-tetrazine by Diels-Alder cycloaddition (Materials and methods). Compared to wild-type GP, GP* and GP*-Cy3/Cy5 maintained approximately 90% and 80% functionality in virus entry, respectively, and neutralization sensitivity to antibody KZ52, which binds an epitope at the base of GP [25]; in an alternative assay, GP* maintained better than 60% infectivity compared with GP (S1 Fig). These analyses indicate that GP*-Cy3/Cy5 maintains near-native functionality and a global prefusion conformation reflective of wild-type GP.

## Low pH and glycan cap cleavage enable GP2 conformational changes

For smFRET imaging, pseudovirions were formed by diluting GP* with an excess of wild-type GP. The GP* protomer was then labeled with Cy3 and Cy5. The labeled virions were immobilized on quartz microscope slides and imaged using total internal reflection fluorescence (TIRF) microscopy (Materials and methods). The ratio of GP to GP* was determined such that only a single fluorophore pair was seen in the majority of virions. We first imaged the conformational dynamics of GPΔmuc at neutral pH in the absence of $Ca^{2+}$, conditions predicted to favor the prefusion conformation. smFRET trajectories acquired from individual GPΔmuc molecules showed a predominant high-FRET state (0.95 ± 0.08), as expected based on the approximately 35-Å distance between the fluorophores estimated through molecular dynamics (MD) simulation of GPΔmuc in the prefusion conformation [25,27,28] (Fig 2B and 2C and S6 Fig). Acidification of the pH to 6, 5.2, and 4.5 led to stepwise reduction in the occupancy of the high-FRET state and increases in the occupancy of a low-FRET state (0.21 ± 0.08; Fig 3A). We

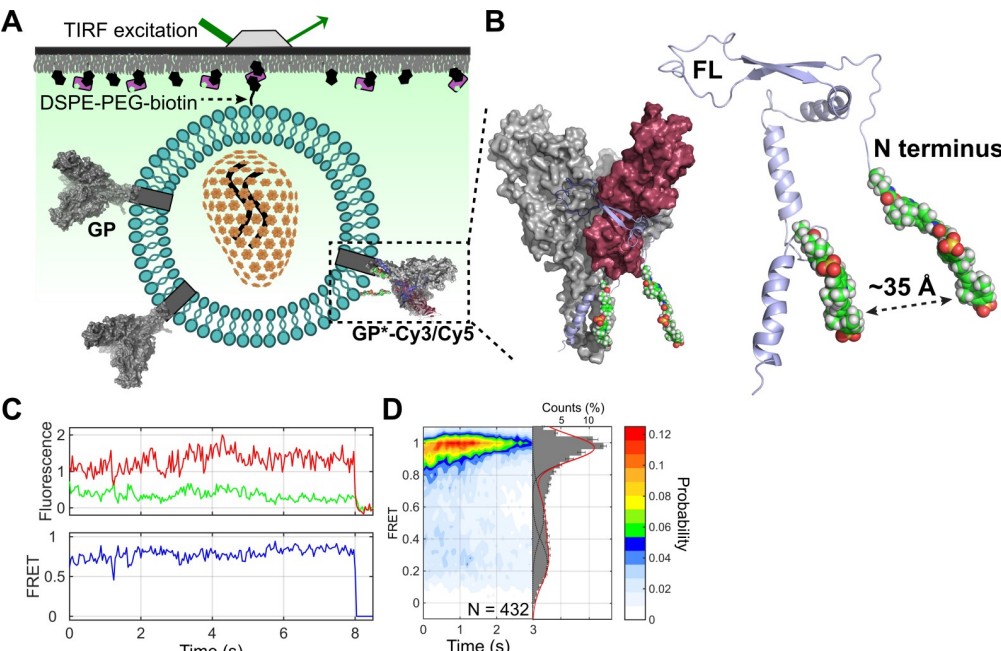

**Fig 2. smFRET imaging assay for visualization of EBOV GP conformational changes related to membrane fusion.**
(A) Experimental setup of smFRET imaging assay. (B) Structural model of the prefusion GPΔmuc trimer with one
protomer labeled with donor and acceptor fluorophores (model based on PDB 5JQ3; Materials and methods).
Unlabeled protomers within the trimer are shown in gray. GP2 (light blue) wraps around GP1 (maroon), with the FL
residing in a hydrophobic cleft at the protomer interface and contacting the neighboring protomer. MD simulation
indicates a time-averaged distance between the fluorophores of approximately 35 Å, consistent with a high-FRET state.
(C) Representative example of fluorescence (donor, green; acceptor, red) and smFRET (blue) trajectories acquired
from an individual GPΔmuc molecule, indicating a predominant high-FRET state at pH 7. Fluorophore
photobleaching occurs at approximately 8 s. (D) Contour plot and FRET histogram indicating the predominant high-
FRET state across the population of GPΔmuc molecules at pH 7. The loss of FRET signal over time shown in the
contour plot arises because of photobleaching of the fluorophores. Overlaid on the FRET histogram are Gaussian fits of
the 3 observed FRET states identified through HMM (Materials and methods). *N* indicates the number of FRET traces
compiled into each contour plot and histogram. EBOV, Ebola virus; FL, fusion loop; FRET, Förster resonance energy
transfer; GP, EBOV envelope glycoprotein; GPΔmuc, GP with the mucin-like domain deleted; HMM, hidden Markov
modeling; MD, molecular dynamics; smFRET, single-molecule Förster resonance energy transfer.

also observed occupancy in an intermediate FRET state (0.52 ± 0.09), which did not respond
to changes in pH. These data indicate that protonation of GP destabilizes the prefusion confor-
mation and promotes conformations in which the GP2 N terminus has moved to positions,
which increase the distance between the 2 fluorophores, giving rise to the low-FRET state. To
test this interpretation, we mutated the furin cleavage site in GP. This gives rise to the
uncleaved precursor, GP0, on which the GP2 N terminus is covalently linked to GP1. GP0 dis-
played stable high FRET even at acidic pH, confirming that the low-FRET state arises from a
conformational change that displaces the GP2 N terminus (S7 Fig).

Glycan cap removal is an essential step during EBOV entry, because it makes available the
NPC1-binding site, and may also facilitate conformational changes in GP related to membrane
fusion [9,32,33]. Therefore, following removal of the glycan cap, we evaluated the conforma-
tional equilibrium of GP$^{CL}$ as a function of pH with smFRET. At pH 7, GP$^{CL}$ predominantly
occupied the high-FRET prefusion conformation as seen for GPΔmuc (Fig 3B). Reduction in
pH decreased the high-FRET occupancy and concomitantly increased the low-FRET occu-
pancy more readily than for GPΔmuc (Fig 3B). Again, intermediate FRET was observed but

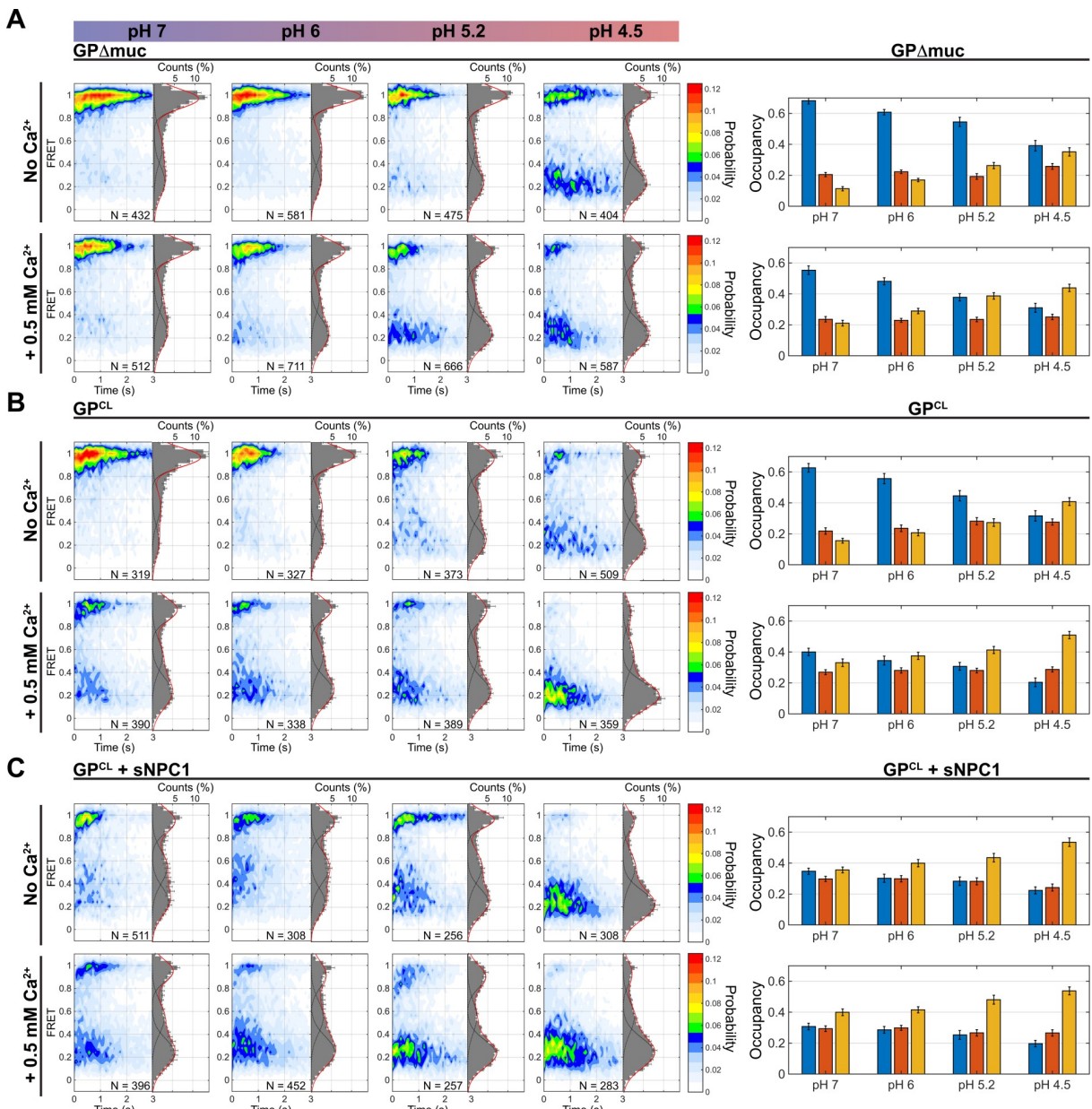

**Fig 3. Ca²⁺, acidic pH, glycan cap removal, and NPC1 binding destabilize the prefusion conformation of GP.** (A) Contour plots displaying the FRET distribution from the population of individual GPΔmuc molecules, at the indicated pH, and in the absence and presence of 0.5 mM CaCl₂. Also shown for each contour plot is the corresponding histogram, overlaid with Gaussian fits for each of the 3 FRET states. *N* indicates the number of FRET traces compiled into each contour plot and histogram. Shown at the right are the occupancies in the high- (blue), intermediate- (orange), and low-FRET (yellow) states, which are displayed as the average of 3 independent groups of traces, with error bars reflecting the standard error. Occupancies in the 3 FRET states are normalized such that their sum equals 100%. (B) The same data for GP^CL and (C) GP^CL bound to sNPC1-C. FRET, Förster resonance energy transfer; GP, EBOV envelope glycoprotein; GPΔmuc, GP with the mucin-like domain deleted; NPC1, Niemann-Pick C1; sNPC1-C, soluble domain C of NPC1.

did not respond to changes in pH. These data suggest that removal of the glycan cap makes GP2 conformation more responsive to changes in pH. Enzyme-linked immunosorbent assays demonstrated that NPC1 binds to GP more readily at acidic pH (S8 Fig). Thus, the acidic pH-induced conformational change in GP may facilitate NPC1 binding.

## Reversibility of GP2 conformational changes

Given that $Ca^{2+}$ accelerated the rate of virus-liposome lipid mixing, we sought to determine if it induced a conformational change in GP2. We therefore imaged GPΔmuc in the presence of 0.5 mM $Ca^{2+}$. Across the pH range tested here, $Ca^{2+}$ destabilized the high-FRET prefusion state of GPΔmuc and increased the occupancy in low FRET (Fig 3A, bottom). For $GP^{CL}$, $Ca^{2+}$ induced a notable decrease in high FRET and an increase in low-FRET occupancy at neutral pH, with the same trend across the pH range (Fig 3B, bottom). For both GPΔmuc and $GP^{CL}$, at pH 7 and 4.5, the effect of $Ca^{2+}$ was reversible; introduction of EDTA to sequester $Ca^{2+}$ returned both GPΔmuc and $GP^{CL}$ to the conformational equilibrium observed prior to addition of $Ca^{2+}$ (S9 Fig).

Consistent with live-cell imaging experiments, virus-liposome lipid mixing required the presence of sNPC1-C [15,16]. We therefore predicted that sNPC1-C binding would induce conformational changes in GP2 related to membrane fusion. To test this prediction, we incubated virions containing $GP^{CL}$ with sNPC1-C prior to smFRET imaging. Even at neutral pH, sNPC1-C induced a dramatic reduction in the occupancy of the high-FRET state and increased occupancy in low FRET (Fig 3C). Reduction in pH decreased occupancy in high FRET. As predicted from the virus-liposome lipid mixing data (Fig 1), the observation of the high-FRET prefusion state was further reduced by the addition of $Ca^{2+}$ across the range of pHs investigated (Fig 3C).

We next asked whether GP2 could return from the low-FRET state after transient exposure to acidic pH. We exposed the virus to pH 4.5 for 10 min before returning to neutral pH. In the absence of $Ca^{2+}$, GPΔmuc fully returned to high FRET after restoration of neutral pH (Fig 4A). Complete reversibility of GPΔmuc was maintained with inclusion of $Ca^{2+}$ (Fig 4A). Thus, the low-FRET state stabilized by acidic pH and $Ca^{2+}$ reflects a reversible conformation of GPΔmuc. In contrast, $GP^{CL}$ only partially returned to predominant high FRET after restoration of neutral pH, and occupancy in an irreversible low-FRET state emerged. The same irreversibility was seen in the presence of $Ca^{2+}$ (Fig 4B). That is, low pH and $Ca^{2+}$ promote transition of $GP^{CL}$ to an irreversible low-FRET state, which is distinct from the reversible low-FRET state seen for GPΔmuc. To test whether restoration of the high-FRET prefusion state correlates with GP's ability to mediate lipid mixing, we exposed virions containing GPΔmuc to pH 4.5 for 10 min followed by restoration of neutral pH and cleavage of the glycan cap. Under these circumstances, $GP^{CL}$ maintained its ability to mediate virus-liposome lipid mixing in the dequenching assay (S2D Fig). The same maintenance of GP-mediated lipid mixing was seen in the presence of $Ca^{2+}$. However, when the glycan cap was removed prior to incubation at pH 4.5 for 10 min, $GP^{CL}$ mediated only a low level of lipid mixing, whether or not $Ca^{2+}$ was present (S2D Fig). Thus, restoration of the prefusion high-FRET state correlates with GP's ability to mediate lipid mixing. Only partial reversibility of the low-FRET state was seen following preincubation with sNPC1-C at pH 4.5 for 10 min. The low-FRET state was entirely irreversible with the additional inclusion of $Ca^{2+}$ (Fig 4C). Thus, incubation with sNPC1-C and $Ca^{2+}$ at acidic pH induced transition of $GP^{CL}$ to a conformation that could not be reversed by restoring neutral pH.

## Discussion

This study provides the first direct observations of conformational changes in intact, trimeric EBOV GP related to membrane fusion. Our results suggest a model of GP-mediated membrane fusion with essential roles for acidic pH, glycan cap removal, NPC1 binding, and endosomal $Ca^{2+}$ (Fig 5). According to this model, prior to membrane fusion, GP has access to minimally 3 conformations. Given the proximity of the fluorophore attachment site at the

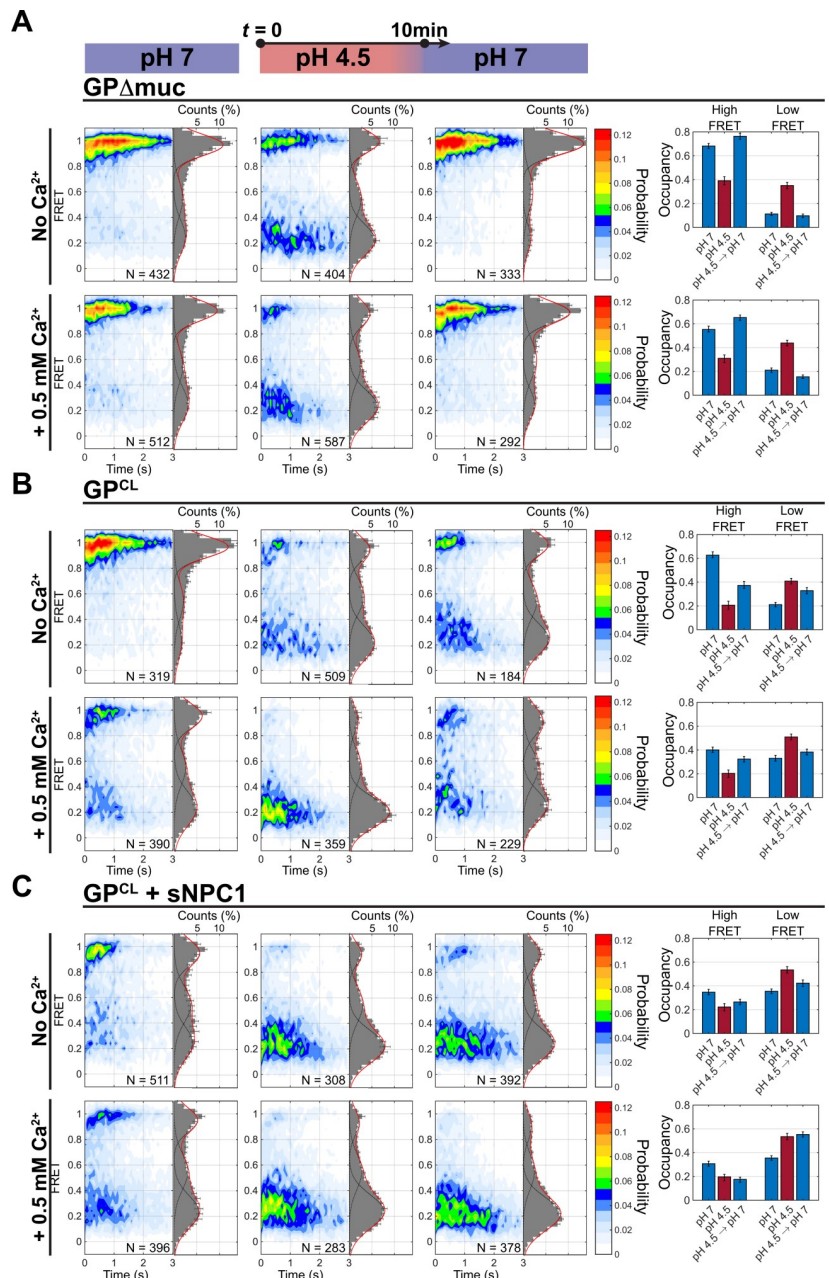

**Fig 4. NPC1 binding induces transition to an irreversible GP conformation.** (A) Contour plots and FRET histograms displaying the FRET distribution across the population of observed GPΔmuc molecules under the indicated conditions, either at pH 7, pH 4.5, or at pH 7 after 10-min exposure to pH 4.5 as indicated by the timeline at the top. *N* indicates the number of FRET traces compiled into each contour plot and histogram. Also shown at the right are the occupancies in high and low FRET under the indicated conditions, which are displayed as the average of 3 independent groups of traces, with error bars reflecting the standard error. (B) The same data for GP<sup>CL</sup> and (C) GP<sup>CL</sup> with sNPC1. FRET, Förster resonance energy transfer; GP, EBOV envelope glycoprotein; GPΔmuc, GP with the mucin-like domain deleted; NPC1, Niemann-Pick C1.

GP2 N terminus to the FL, the FRET states observed here likely reflect conformations in which the FL adopts distinct positions. First, the high-FRET state is consistent with the prefusion conformation that has been described through x-ray crystallography [25,27]. Second, the

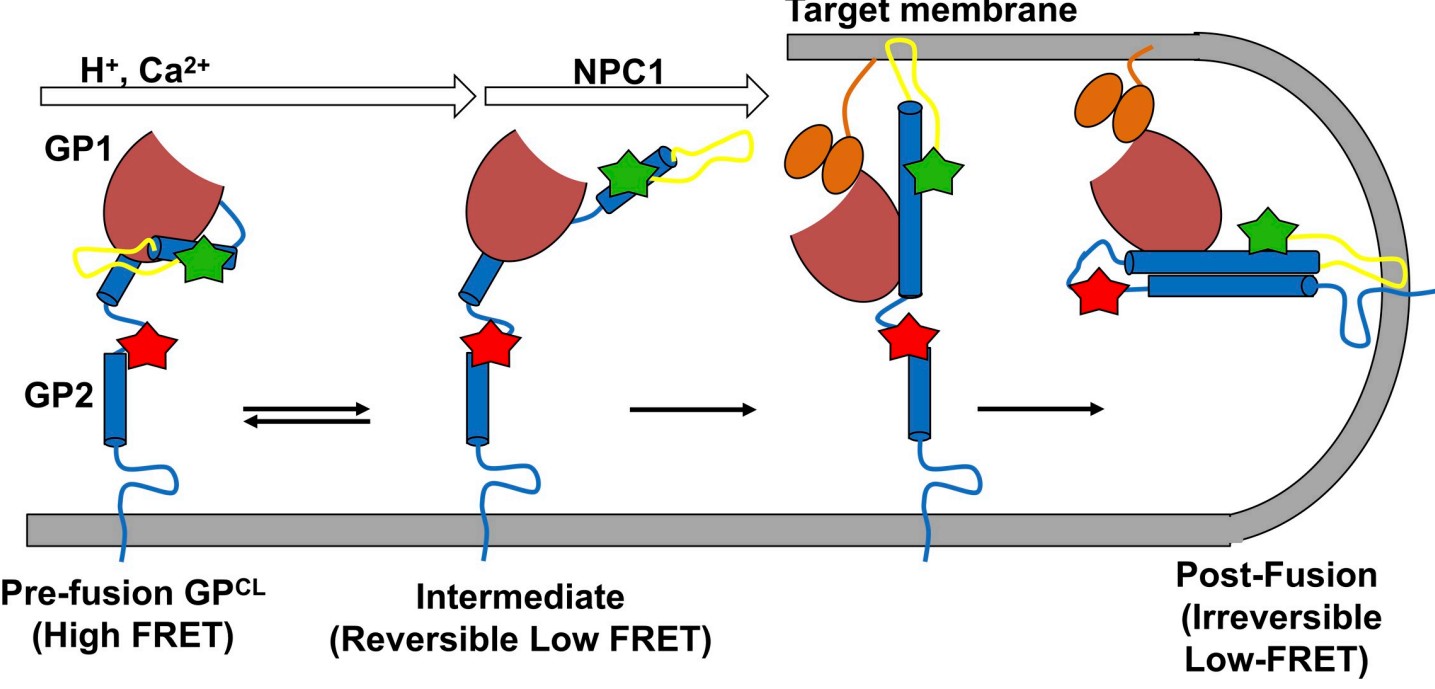

**Fig 5. Mechanistic model of GP-mediated membrane fusion.** The present observations are consistent with a model in which acidic pH and $Ca^{2+}$ shift the prefusion conformational equilibrium of GP in favor of a conformation optimal for NPC1 binding. In this conformation, the GP2 N terminus and FL have moved out of the prefusion cleft to a position distal to the viral membrane. NPC1 then triggers irreversible transition to the postfusion conformation necessary for virus-liposome lipid mixing. The intermediate FRET state may represent an additional on-pathway intermediate or a conformation not relevant to the mechanism of viral fusion. FL, fusion loop; FRET, Förster resonance energy transfer; GP, EBOV envelope glycoprotein; NPC1, Niemann-Pick C1.

intermediate FRET state may reflect a localized rearrangement of the FL that precedes the putative large-scale displacement predicted by the canonical model of class-I viral membrane fusion [34]. Alternatively, because the stability of the intermediate FRET state is not affected by any of the variables considered in this study, we cannot rule out the possibility that it reflects an off-pathway conformation that is not relevant to the mechanism of membrane fusion. Third, prefusion GP has access to a reversible low-FRET state in which the GP2 N terminus and the FL have moved away from the trimer axis or the viral membrane. Our observations demonstrate that acidic pH and $Ca^{2+}$ directly initiate conformational remodeling of GP by shifting the equilibrium in favor of the reversible low-FRET conformation. Importantly, prior to removal of the glycan cap, the high-FRET prefusion conformation is reformed upon restoration of neutral pH or removal of $Ca^{2+}$. We anticipate the transition to the postfusion 6HB would be irreversible because of the thermodynamic stability of similar envelope GPs in this conformation [35]. Thus, $Ca^{2+}$ and acidic pH alone do not efficiently promote transition of GPΔmuc to the postfusion 6HB conformation on the timescale of our measurements. Rather, we speculate that the reversible low-FRET state is an intermediate conformation that is on pathway to the postfusion 6HB. This conformation may incorporate an FL position in which it can interact with the target membrane. Stabilization of an on-pathway intermediate conformation may indicate a mechanism by which acidic pH and $Ca^{2+}$ promote virus-liposome lipid mixing.

In addition to mediating the conformational equilibrium of prefusion GP, the acidic pH of the endosome activates cellular cathepsin proteases. The cathepsins remove the mucin-like and glycan cap domains from GP1, exposing the receptor-binding site [9]. Our results support a critical role for the glycan cap beyond obscuring the receptor-binding site. Cleavage of the

glycan cap made GP more responsive to reduction in pH, facilitating transition of GP2 to low-FRET conformations. A parallel smFRET imaging study of GP with fluorophores attached to the N termini of GP1 and GP2 reported a conformational change in GP1 at neutral pH upon glycan cap cleavage [36]. Taken together, these studies demonstrate that glycan cap cleavage induces a change in GP1, which in turn enables flexibility in GP2. Upon acidification, the GP2 N terminus and FL have greater ability to adopt positions outside the hydrophobic cleft in the neighboring protomer. In particular, removal of the glycan cap enables GP2 to transition to a low-FRET conformation that is not reversible after restoration of neutral pH. Thus, the glycan cap assists with allosterically regulating the timing of GP2 conformational changes, ensuring that GP does not prematurely trigger prior to arrival of the virion in a late endosome that contains the cathepsins and NPC1. This proposed mechanism is reminiscent of the prM protein in Dengue virus, which prevents triggering of the E GP prior to proteolytic cleavage of prM [37].

Glycan cap removal also has the well-established role of permitting GP binding to the NPC1 receptor, which is an essential step in EBOV entry [9–11,15,16]. Consistent with these observations, we find NPC1 binding is essential for GP-mediated lipid mixing. Our smFRET data provide the additional insight that NPC1 promotes GP2 transitioning from the reversible low-FRET conformation to the irreversible low-FRET conformation. The NPC1-binding site in GP1 is separated from the FL in GP2 by approximately 20Å, which indicates an allosteric connection between these functional centers. We documented a similar connection between the receptor-binding site and the fusion peptide in a study of influenza HA [24].

Although the site of $Ca^{2+}$ binding to intact GP is currently unknown, one possibility is direct interaction with the FL, as previously demonstrated in a study of the isolated FL [20]. Sequence analysis of EBOV GP, as well as other filoviral GPs, indicates universally conserved acidic residues in the FL (D521, E522, and E540). Coordinating $Ca^{2+}$ may neutralize the negative charge of these residues, triggering efficient release of the FL from the hydrophobic cleft. This potential charge neutralization may also facilitate FL insertion into the target membrane, as described for the Rubella virus (RuV) E1 GP [38]. $Ca^{2+}$ may also stabilize a local conformation of the FL that is optimal for insertion into the target membrane. RuV E1 and EBOV GP now provide 2 examples of viral GPs that sense cellular $Ca^{2+}$ as a means of regulating the timing and efficiency of fusion, analogously to cellular synaptotagmin, which promotes synaptic vesicle fusion [39].

EBOV experiences relatively high $Ca^{2+}$ concentration in the extracellular space (approximately 1–2 mM). Following internalization via endocytosis the $Ca^{2+}$ concentration decreases to approximately μM because of the action of pH-dependent $Ca^{2+}$ channels [40]. Estimates of the $Ca^{2+}$ concentration further along the endocytic pathway vary widely (3 to 600 μM) [23,40,41]. Previous studies have documented the importance for EBOV entry of TPC channels [17], which release $Ca^{2+}$ from the late endosome [14]. Inhibition of TPC function prevented EBOV infection. Likewise, SERMs perturb the outflow of $Ca^{2+}$ from the late endosome, which may explain their anti-EBOV activity [18,19]. Removal of $Ca^{2+}$ from the endosome through chelation also reduced GP-mediated EBOV entry [19]. These studies clearly implicate endosomal $Ca^{2+}$ as an important regulator of EBOV entry. We observed that maximal GP-mediated lipid mixing occurred over a range of 0.1 to 0.5 mM $Ca^{2+}$, whereas excess $Ca^{2+}$ (>1 mM) limited the extent of lipid mixing. One possible explanation is that elevated $Ca^{2+}$ concentration inactivates GP because of premature triggering prior to efficient engagement with NPC1 or transition to a conformation that is not competent for fusion. In this case, the presence of the glycan cap on GP prior to internalization into the endosome may prevent against triggering in the $Ca^{2+}$-rich extracellular space. Alternatively, elevated $Ca^{2+}$ concentration may inhibit conformational changes in GP related to fusion.

As summarized in Fig 5, our data are consistent with a model in which acidic pH and $Ca^{2+}$ shift the conformational equilibrium toward states in which the GP2 N terminus and the FL move out of their positions in the prefusion conformation. These conformational changes remain reversible until the glycan cap is removed. NPC1 binding then drives GP into the irreversible low-FRET conformation. This conformation may reflect the post fusion 6HB or a conformation preceding the 6HB in the fusion pathway. The fluorescence dequenching data presented here cannot distinguish between hemifusion and fusion pore formation. Furthermore, not all hemifusion events proceed to fusion pore formation [33,34]. Thus, the current observations do not specify whether acidic pH, $Ca^{2+}$, and NPC1 binding are sufficient to trigger GP-mediated membrane fusion or whether additional factors are necessary [15]. Whatever the identity of the irreversible low-FRET state, maintaining reversibility in GP conformation is correlated with the ability of GP to mediate lipid mixing. This observation highlights the importance of the putative role of the glycan cap in preventing premature transition to the irreversible low-FRET state. Future studies are necessary to generate a mechanistic understanding of how the glycan cap and NPC1 binding in GP1 mediate the timing of GP2 conformational events. Likewise, specification of the mechanism by which $Ca^{2+}$ can be both an accelerant and an inhibitor of GP-mediated fusion will require further investigation. The smFRET imaging approach presented will be a valuable tool in elucidating these questions.

# Materials and methods

## Production of pseudovirions with EBOV GP and the HIV-1 core

Pseudotyped virions were produced by transfecting HEK293T cells with plasmids encoding EBOV GPΔmuc from the Mayinga strain and HIV-1 GagPol. Cells were maintained in DMEM (Thermo Fisher, Waltham, MA) supplemented with 10% FBS (Gemini Bio, West Sacramento, CA), 100 U/mL penicillin/streptomycin (Thermo Fisher, Waltham, MA), and 2 mM L-glutamine (Thermo Fisher, Waltham, MA). At 12 h post transfection, the medium was replaced. Viral supernatant was harvested at 48 h post transfection, passed through a 0.45-μm filter to remove cell debris, and concentrated 10-fold by centrifugation for 2 h at 20,000*g* over a 10% sucrose cushion. The virus particles were then shown to be positive for EBOV GP and HIV-1 p24 by Western blot. The GPΔmuc contained on the virion surface was >90% proteolytically cleaved into GP1 and GP2. To produce virions with $GP^{CL}$, the particles were incubated with thermolysin (0.2 mg/mL) from *Bacillus thermoproteolyticus* (Roche, Basel, Switzerland) at 37 ºC for 1 h, as described by Miller and colleagues [9]. Virions containing GP0 were produced in the same way after mutating the native furin cleavage site in GPΔmuc (RRTRR) to GGGGG by overlap extension PCR, which abolished cleavage of GP1 and GP2.

## Noncanonical amino acid incorporation into EBOV GP*

We incorporated 2 TCO* ncAAs into the GP2 domain of EBOV GPΔmuc through amber stop codon suppression [31] (S4 and S5 Figs). To this end, we introduced TAG codons at position 501 of GP2 immediately after the furin cleavage site and at position 610 after the $CX_6CC$ motif by site-directed mutagenesis. Translation proceeds through the UAG codons on the mRNA only in the presence of an orthogonal tRNA ($tRNA^{Pyl}$), which recognizes the UAG codon, and a corresponding aminoacyl-tRNA synthetase ($NESPylRS^{AF}$). $NESPylRS^{AF}$ aminoacylates the suppressor tRNA with TCO*, facilitating its incorporation at specific locations into GP, forming GP*. The efficiency of amber suppression is limited because of competition of the eukaryotic release factor 1 (eRF1) with $tRNA^{Pyl}$ [42]. Expression of the dominant negative eRF1 E55D mutant increased amber suppression efficiency.

Readthrough of the amber stop codons was evaluated in the context of the GP0 mutant so that detection of truncation at positions proximal to the fusion peptide were not complicated by the presence of GP1. The translation of full-length EBOV GP0 was first evaluated by transfecting HEK293T cells with plasmids encoding the TAG-mutated GP0 and NESPylRS$^{AF}$/tRNA$^{Pyl}$, eRF1 E55D, and HIV-1 Gag-Pol. The growth medium was supplemented with 0.5 mM TCO* ncAA (SiChem, Bremen, Germany). At 48 h post transfection, the virus was harvested, concentrated by centrifugation, and evaluated by Western blot (S5 Fig). In the presence of UAG codons at both positions 501 and 610, full-length GP0* was translated at approximately 5% to 10% efficiency as compared to wild-type GP and was dependent on the presence of NESPylRS$^{AF}$/tRNA$^{Pyl}$, TCO*, and eRF1 (E55D). Uncropped blots are provided in S1 Raw Images.

## Pseudovirus fusion and infectivity assays

The function of the GP variants in membrane fusion and virus entry were evaluated in 2 ways. First, we specifically tested functionality in membrane fusion using a β-lactamase (Blam)-based enzymatic assay (LiveBlazer FRET B/G kit, Thermo Fisher, Waltham, MA) [43]. Virions with either GP or GP* on their surface were formed as described above with an additional plasmid encoding Blam fused the HIV-1 Vpr protein. Virions with GP*-Cy3/Cy5 were formed by labeling GP* as described below. Of note, for the purpose of evaluating the functionality of GP* and GP*-Cy3-Cy5, pseudovirions were not formed with excess wild-type GP as done for smFRET imaging experiments. Virions were harvested and concentrated by centrifugation, as described above. Virus pellets were resuspended in RPMI medium (Thermo Fisher, Waltham, MA; supplemented with 0.2% BSA, 10 mM Hepes [pH 7]) and incubated with prechilled Vero cells for 1 h in a 96-well plate on ice to facilitate virus attachment to the cell surface. The plate was centrifuged for 30 min at 3,700 rpms at 4˚C. The unbound virus was removed by washing the cells with Ca$^{2+}$-free DMEM. The cells were incubated at 37˚C for 90 min to permit viral entry in Ca$^{2+}$-free complete DMEM, supplemented with 10 mM HEPES (pH 7), 0.2% BSA, and in the absence or presence of 0.5 mM CaCl$_2$. The cells were then incubated in complete DMEM media (with Ca$^{2+}$) containing 20 mM NH$_4$Cl for 90 min at 37˚C.

The cells were then loaded with the CCF4-AM fluorophore in the presence of 250 mM probenecid at 11˚C overnight, and the infection was monitored by detecting cleavage of the CCF4-AM by Blam using a Synergy HT plate reader (Biotek, Winooski, VT). Virus particles containing GP* demonstrated fusion at approximately 90% of the efficiency of wild-type GP. Alternatively, virus particles were incubated with 50 μg/mL KZ52 antibody prior to introduction to the Vero cells, which resulted in neutralization of the GP*-containing virus as efficiently as that containing wild-type GP.

In the second assay, GP was pseudotyped onto a vesicular stomatitis virus (VSV) core containing a GFP expression reporter, as described by Whitt [44]. The 293Ts were transfected with GP using PEI transfection reagent. Transfected 293Ts were then infected with VSV-ΔG-GFP viral stocks at an MOI of 3. Virus-containing supernatant was collected 24 h post infection, passed through a 0.45 um filter, aliquoted, and frozen at −80˚C. Thawed aliquots were either labeled or mock labeled with DiD as described below for DiO and used to infect naïve 293Ts in 6-well dishes for 5 h. DiD was used to evaluate the effects of a lipophilic fluorophore because it is spectrally distinct from GFP and chemically similar to DiO, which was used in the virus-liposome lipid mixing assay. Cells were collected and resuspended in PBS on ice and evaluated for GFP expression using flow cytometry.

Finally, to evaluate GP-mediated virus entry in the presence of inhibitory compounds, stocks of eGFP expressing HIV virions pseudotyped with Ebola GP were produced as

previously detailed by Markosyan and colleagues [37]. Briefly, $6 \times 10^5$ HEK293 cells were seeded in 6-well plates and transfected using TransIT-LT1 with the following plasmids 24 h later: 1250 ng of the GFP-expressing lentiviral transfer vector pNL-EGFP/CMV-WPREΔU3 [42] (Addgene 17579, Watertown, MA), 930 ng of the self-inactivating second-generation lentiviral packaging vector pCD/NL-BH*ΔΔΔ [43], and 310 ng of the GP expression plasmid [37] (Addgene 86021). For infections, U2OS cells were treated for 2 h with the indicated concentration of verapamil, enclomiphene, tetrandrine, or DMSO (Sigma-Aldrich, St. Louis, MO). Following that, virus was added, and cells were incubated overnight in the presence of both virus and drug. The following morning media was replaced with fresh DMEM and incubated for an additional 48 h at which time cells were harvested and GFP expression was assessed by flow cytometry. Alternatively, virus and tetrandrine were incubated at room temperature for 30 min before incubating with target U2OS cells for 2 h on ice, after which virus was removed and cells were cultured for 72 h in fresh DMEM. Percent GFP positive cells in the drug-treated samples were compared to that observed with DMSO treatment, and inhibition curves were generated in Prism 8 (GraphPad Software, San Diego, CA). All numeric infectivity and fusion data are provided in S1 Data.

### Virus-liposome lipid mixing assay

The in vitro assay used to probe the lipid mixing of virions and liposomes was described previously for a study of HA [24]. Briefly, virions with either GP$^{CL}$ or GPΔmuc were incubated with 50 μM DiO for 2 h at room temperature to label viral membrane. The labeled virions were purified by density gradient ultracentrifugation on a 6% to 30% Optiprep (Sigma-Aldrich, St. Louis, MO) gradient at 35,000 rpms for 1 h. Separately, liposomes were formed with a composition of 4:4:2:0.5:0.1 ratio of 1,2-dioleoyl-sn-glycero-3-phosphocholine (DOPC; Avanti Polar Lipids, Alabaster, AL), 1-palmitoyl-2-oleoyl-glycero-3-phosphocholine (POPC; Avanti Polar Lipids, Alabaster, AL), cholesterol (Avanti Polar Lipids, Alabaster, AL), phosphatidyl serine (Avanti Polar Lipids, Alabaster, AL), and Ni-NTA DGS lipid (Avanti Polar Lipids, Alabaster, AL). Lipids were dissolved in chloroform and were evaporated under a stream of Ar gas. The dried lipid film was then suspended in phosphate buffer (pH 7.5) to a final concentration of 10 mg/mL. The liposomes were extruded through a polycarbonate membrane filter with pore sizes of 200 nm. To coat the liposome with the receptor, liposomes were incubated with polyhistidine-tagged sNPC1-C. The Ni-NTA DGS in the liposome was present in 100-fold molar excess over sNPC1-C to ensure minimal unbound sNPC1-C.

Liposomes (with or without sNPC1-C) were combined with DiO-labeled virions containing either GP$^{CL}$ or GPΔmuc and incubated at room temperature for 5 min at pH 7.5. Buffer was then added by stopped-flow to adjust the pH to the desired level, followed by time-based measurement of DiO fluorescence. For measuring the effect of $Ca^{2+}$ on dequenching kinetics, $CaCl_2$ was added to the virus-liposome mixture at the desired concentration. DiO was excited at 450 nm, and fluorescence was detected at 525 nm at 1 s intervals for 300 s at room temperature in a QuantaMaster 400 fluorimeter (Horiba, Kyoto, Japan). Dequenching data were fit to the exponential function $A(1-e^{-kt})$ using a least-squares algorithm in Matlab (MathWorks, Natick, MA). All numeric lipid mixing data is provided in S1 Data.

### Expression and purification of sNPC1-C

sNPC1-C containing an N-terminal polyhistidin tag was expressed by transfection of HEK293 Freestyle cells (Thermo Fisher, Waltham, MA) with polyethyleneimine (PEI MAX, Polysciences, Warrington, PA), as described by Miller and colleagues [9]. Six days post transfection, cell culture supernatant containing secreted sNPC1-C was harvested. sNPC1-C was

extracted from the supernatant by passage over Ni-NTA agarose beads (Thermo Fisher, Waltham, MA). The sNPC1-C-bound Ni-NTA beads were washed and eluted by standard means. Purified sNPC1-C was dialyzed into 20 mM Tris-HCl, 100 mM NaCl, 2 mM β-mercaptoethanol, and 10% glycerol, followed by concentration with Vivaspin 6 spin filters (Sartorius, Göttingen, Germany). The purified sNPC1-C was stored at −80˚C until to use.

## Fluorescently labeling EBOV GP* for smFRET imaging

For smFRET imaging, virions were formed with on average a single GP* protomer among the distribution of wild-type GP, as has been done previously for smFRET imaging studies of influenza HA and HIV-1 Env [24,45]. Briefly, HEK293T cells were transfected with a 1:5 ratio of plasmids encoding GP* and wild-type GP, and plasmids encoding HIV-1 Gag-Pol, NESPylRS$^{AF}$/tRNA$^{Pyl}$, and eRF1 (E55D), and grown in the presence of 0.5 mM TCO* ncAA. Given the limited efficiency (approximately 10%) of reading through 2 stop codons during translation of GP*, the ratio of GP* to wild-type GP plasmid ratio equates to an approximately 50-fold excess of wild-type GP protein over GP* protein. Virus was harvested, concentrated by centrifugation, and resuspended in phosphate buffer (pH 7.5). Virions were then labeled by incubation with 2 mM 3-(p-benzylamino)-1,2,4,5-tetrazine-Cy3 and 2 mM 3-(p-benzylamino)-1,2,4,5-tetrazine-Cy5 for 10 minutes at 37˚C (Jena Bioscience, Jena, Germany). Because we introduced the same ncAA at both positions 501 and 610 of GP2, 50% of the labeled virions contained either 2 Cy3 fluorophores or 2 Cy5 fluorophores, which were easily removed from consideration during analysis (see below). Following the labeling reaction, DSPE-PEG$^{2000}$-biotin (Avanti Polar Lipids, Alabaster, AL) was added and incubated with the virions for 30 min at a final concentration of 6 μM (0.02 mg/mL). The labeled virus was purified from unbound dye and lipid by ultracentrifugation for 1 h at 35,000 rpm over a 6% to 30% Optiprep (Sigma-Aldrich, St. Louis, MO) gradient in 50 mM Tris (pH 7.4), 100 mM NaCl. The gradient was fractionated, and the fractions containing virus were identified by p24 Western blot. The purified fluorescently labeled virus was stored at −80˚C for use in smFRET imaging experiments.

## smFRET imaging assay with TIRF microscopy

smFRET imaging experiments were performed on a custom-built prism-based TIRF microscope as described previously by Das and colleagues [24]. The fluorescently labeled virions were immobilized on the polyethylene glycol (PEG)-passivated, streptavidin-coated quartz microscope slides, and mounted on an XY automated stage (Applied Scientific Instrumentation, Eugene, OR). The surface-bound virions were illuminated by the evanescent field generated by total internal reflection of a 532-nm wavelength laser (Coherent, Santa Clara, CA) at an intensity of 0.2 kW/cm$^2$. Fluorescence emission was collected through a 1.2-NA 60X water-immersion objective (Olympus, Tokyo, Japan), and passed through a T550LPXR dichroic filter (Chroma) to remove scattered laser light. Donor and acceptor emission were separated with a T635LPXR dichroic filter (Chroma, Bellows Falls, VT) and imaged on 2 parallel ORCA-Flash4.0-V2 sCMOS cameras (Hamamatsu, Hamamatsu City, Japan). Images were collected at 25 frames/second using MicroManager software (micro-manager.org).

During imaging buffer containing either 50 mM sodium phosphate pH 6 to 7 or sodium acetate pH 4.5 to 5.2 and 50 mM NaCl were introduced to the surface-bound virions using a syringe pump (Harvard Apparatus, Holliston, MA). For imaging the effects of Ca$^{2+}$ on GP, conformation 0.5 mM CaCl$_2$ was included in the imaging buffer. Also contained in the imaging buffer was a cocktail of triplet-state quenchers (1 mM trolox, 1 mM cyclooctatetraene, 1 mM nitrobenzyl alcohol), and an enzymatic system for removal of molecular oxygen, which

included 2 mM protocatechuic acid and 8 nM protocatechuate 3,4-deoxygenase. All smFRET imaging experiments were performed at room temperature.

## Data analysis

smFRET data analysis was performed using the SPARTAN software package with additional custom-written scripts in Matlab (MathWorks, Natick, MA) [46] (https://www.scottcblan chardlab.com/software). smFRET trajectories were automatically identified according to the selection criteria: (1) FRET between donor and acceptor was detectable for minimally 15 frames before photobleaching; (2) donor and acceptor fluorescence traces displayed a single step photobleaching event, which indicates the presence of a single donor fluorophore and a single acceptor fluorophore per virion; (3) the correlation coefficient of donor and acceptor fluorescence traces was less than −0.1; (4) the signal-to-noise ratio of the total fluorescence, defined as the ratio of the magnitude of the photobleaching event to the variance of the background signal, was greater than 8. Selected smFRET trajectories that met these criteria were fit to a hidden Markov model (HMM) consisting of 4 states, with FRET values of 0.00 ± 0.06, 0.21 ± 0.08, 0.52 ± 0.08, and 0.95 ± 0.08 (mean ± standard deviation) using the segmental $k$-means algorithm. HMM analysis assigns each FRET data point to one of the 4 states, which permits calculation of the time-averaged occupancies of the states. All the observed FRET data points prior to photobleaching were compiled into time-resolved histograms and displayed as contour plots. Data points assigned to the 0-FRET state were not plotted to enable clearer visualization of the 0.21-FRET state. Thus, the loss of a FRET signal over time in the contour plots results from photobleaching of the fluorophores. The time-resolved histograms were integrated over the observation window to yield one-dimensional FRET histograms, which were normalized such that the total probability, summed across all bins equals one. The error in the probability per histogram bin was determined by randomly assigning each FRET trajectory to 1 of 3 different groups and calculating the standard error of the mean of those groups. The explicit probabilities per histogram bin are shown in S1 Data.

## MD simulations

An atomic model of the EBOV GP ectodomain was generated using coordinates determined with x-ray crystallography [27] (PDB 5JQ3). Models of the Cy3- and Cy5-tetrazine fluorophores and the TCO* ncAA were previously described by Das and colleagues [24]. In addition, the transition dipoles of the fluorophores were calculated using the CIS method in Gaussian 9. Fluorophores were attached to GP, hydrogen atoms were added to the protein, and the entire system was solvated in explicit water with charge-neutralizing ions in the LEaP module of AmberTools. The system was parameterized with the amber force field (ff14SB), the TIP3P water model, and the Generalized amber Force Field (GAFF2). Simulations were run at the Tufts High Performance Computing Center and on the c3ddb cluster at the Massachusetts Green High-Performance Computing Center using NAMD version 2.10. The system was energy minimized for 0.1 ns, followed by a 50 ns simulation run in the NPT ensemble, with temperature and pressure maintained at 300 K and 1 atm through the use of Langevin dynamics and the Nosé-Hoover Langevin piston method, respectively. The centers of mass of the Cy3 and Cy5 fluorophores were used to calculate the distance between the fluorophores at each frame of the simulation in VMD. The $\kappa^2$ value, which parameterizes the rotational orientation of the 2 fluorophores, was calculated according to $\kappa^2 = [\cos \theta_{DA} - 3\cos \theta_D \cos \theta_A]^2$, where $\theta_{DA}$ is the angle between the transition dipoles of the donor and acceptor fluorophores, and $\theta_D$ and $\theta_A$ are the angles between the donor and acceptor dipoles, respectively, and the vector that separates the 2 fluorophores [47].

## Supporting information

**S1 Fig. Functional evaluation of modified pseudovirions containing EBOV GP.** (A) Infectivity of GFP-expressing VSV particles pseudotyped with GP or GP* or labeled with the lipophilic fluorophore DiD. DiD was used in the infectivity measurements to evaluate the effect of a lipophilic fluorophore because it is spectrally distinct from GFP and chemically similar to the DiO fluorophore used in the lipid mixing assay. DiO excitation overlaps with that of GFP and thus could not be used in this assay. Infectivity is normalized to wild-type GP with an unlabeled virion. (B) Viral fusion was measured using a Blam-based virus entry assay (Materials and methods). HIV pseudovirions containing either GP, GP*, or GP*-Cy3/Cy5, and in the absence or presence of antibody KZ52. For the purpose of evaluating the functionality of GP* and GP*-Cy3-Cy5, pseudovirions were not formed with excess wild-type GP as done for smFRET imaging experiments. Pseudovirions were preincubated with 0.5 mM $CaCl_2$ where indicated. Some enhancement of fusion was observed following the preincubation with $CaCl_2$ indicating that the endosomal $Ca^{2+}$ concentration in the target cells used here may be not be optimal for GP-mediated fusion. All GP variants were efficiently inhibited by KZ52, confirming that the TCO* residues do not perturb the global conformation of GP. Data are presented as the average of 3 independent measurements, with error bars reflecting the standard deviation. EBOV, Ebola virus; GFP, green fluorescent protein; GP, EBOV envelope glycoprotein; HIV, human immunodeficiency virus; smFRET, single-molecule Förster resonance energy transfer; VSV, vesicular stomatitis virus.
(TIF)

**S2 Fig. Proteolytic cleavage and NPC1 binding, and physiological $Ca^{2+}$ are required for virus-liposome lipid mixing.** (A) As described in Fig 1, fluorescently labeled GP-containing virions were incubated with liposomes at the indicated pH and in the absence (red) or presence (blue) of $Ca^{2+}$. No fluorescence dequenching was observed in the presence of NPC1 prior to removal of the glycan cap, regardless of the presence of $Ca^{2+}$ or the pH. (B) In contrast to the acceleration observed in the presence of $Ca^{2+}$, only slow dequenching was seen at pH 5.2 with NPC1 in the presence of 0.5 mM $MgCl_2$ or $ZnCl_2$. (C) Equivalent levels of dequenching were observed across a range of physiological $Ca^{2+}$ concentrations (0.1–0.5 mM) at pH 5.2 with NPC1. However, $Ca^{2+}$ concentrations of at least 1 mM led to a loss of dequenching. (D, top) Dequenching observed for pseudovirions containing GPΔmuc that were incubated at pH 4.5 for 10 min, followed by removal of the glycan cap with thermolysin. (Bottom) Dequenching observed for pseudovirions that were incubated after glycan cap removal. The same data are presented in the absence and presence of $Ca^{2+}$, as indicated. In all panels, data are presented as the percentage of maximal dequenching seen upon addition of 1% triton X-100 and as an average of 3 independent measurements, with error bars reflecting the standard deviation. GP, EBOV envelope glycoprotein; GPΔmuc, GP with the mucin-like domain deleted; NPC1, Niemann-Pick C1.
(TIF)

**S3 Fig. $Ca^{2+}$ channel antagonists and a selective estrogen receptor modulator inhibit GP-mediated virus entry.** Pseudovirus infectivity was tested in the presence of compounds known to affect endosomal $Ca^{2+}$ (Materials and methods). Infectivity is presented as a percentage of DMSO control. Data are presented as the average of 3 to 6 independent measurements, with error bars reflecting the standard errors. GP, EBOV envelope glycoprotein.
(TIF)

**S4 Fig. Experimental protocol for formation of pseudovirions containing a single GP* protomer.** See Materials and methods for details. GP, EBOV envelope glycoprotein.
(TIF)

**S5 Fig. Read through of 2 amber stop codons leads to generation of full-length GPΔmuc.**
(A) Wild-type GPΔmuc and GP0 efficiently incorporate into virions formed with the HIV core. Western blot demonstrating efficient furin-mediated cleavage of GP0. The HIV capsid protein, p24, serves as a loading control. (B) Translation of full-length GP* requires the presence of NESPylRS$^{AF}$/tRNA$^{Pyl}$, the TCO* ncAA, and the dominant negative E55D mutant of eRF1. eRF1, eukaryotic release factor 1; GP, EBOV envelope glycoprotein; GPΔmuc, GP with the mucin-like domain deleted; HIV, human immunodeficiency virus; TCO*, *trans*-cyclooct-2-ene-L-lysine.
(TIF)

**S6 Fig. MD simulation of fluorophore dynamics indicates an average interfluorophore distance of approximately 33 Å.** A molecular model of GP*-Cy3-Cy5 was generated based on the prefusion structure of GP (PDB accession code 5JQ3; Materials and methods). The histogram of distances between the centers of mass of the 2 fluorophores is shown. The average interfluorophore distance was 35 ± 4 Å, consistent with the 0.95-FRET state that predominates for GPΔmuc at pH 7. Models of the Cy3 and Cy5 fluorophores are shown with the calculated transition dipoles used to calculate the orientation factor $\kappa^2$, which is 2/3 for freely tumbling fluorophores. The fluorophore orientations determined in the 50-ns trajectory led to $\kappa^2 = 0.645 \pm 0.320$, indicating that on average the fluorophores are freely tumbling. Their relative rotational orientations vary considerably over time scales that are far shorter than our experimental time resolution of 40 ms. FRET, Förster resonance energy transfer; GP, EBOV envelope glycoprotein; GPΔmuc, GP with the mucin-like domain deleted; MD, molecular dynamics.
(TIF)

**S7 Fig. The uncleaved precursor GP0 does not transition to the low-FRET state at acidic pH.** Contour plots and FRET histograms displaying the FRET distribution from the population of individual GP0 molecules at the indicated pH. Histograms are overlaid with Gaussian distributions as in Figs 2–4. *N* indicates the number of FRET traces compiled into each contour plot and histogram. FRET, Förster resonance energy transfer; GP, EBOV envelope glycoprotein.
(TIF)

**S8 Fig. Acidic pH increases the extent of NPC1 binding to GP.** Pseudovirions containing GP$^{CL}$ were incubated with FLAG-tagged sNPC1-C across a range of pHs. Following incubation, excess sNPC1-C was removed with neutral pH buffer. The extent of binding was determined in an ELISA assay using an anti-FLAG antibody conjugated to horseradish peroxidase (Materials and methods). Greater binding is seen at acidic pH, which may be due to acidic pH facilitating transition of GP to a conformation optimal for NPC1 binding. Data are presented as the average of 3 independent measurements, with error bars reflecting the standard error. ELISA, enzyme-linked immunosorbent assay; GP, EBOV envelope glycoprotein; NPC1, Niemann-Pick C1; sNPC1-C, soluble domain C of NPC1.
(TIF)

**S9 Fig. EDTA reverses the conformational change induced by Ca$^{2+}$.** (A) Contour plots and FRET histograms for GPΔmuc and GP$^{CL}$ acquired under the indicated conditions. FRET data are displayed as in Figs 2–4. *N* indicates the number of FRET traces compiled into each contour plot and histogram. (B) Occupancies in the high- (blue), intermediate- (orange), and low-FRET (yellow) states determined through HMM analysis. Occupancies in the 3 FRET states are normalized such that their sum equals 100%. EDTA, ethylenediaminetetraacetic acid;

FRET, Förster resonance energy transfer; GP, EBOV envelope glycoprotein; GPΔmuc, GP with the mucin-like domain deleted; HMM, hidden Markov modeling.
(TIF)

**S1 Data. Numeric data for each figure panel.**
(PDF)

**S1 Raw Image. Original images for all Western blots.**
(TIF)

# Acknowledgments

The authors wish to thank Angela Howard and Ramesh Govindan for critical reading of this manuscript, Dr. Edward Lemke for providing the NESPylRS[AF]/tRNA[Pyl] and eRF1 plasmids, and Dr. Dennis Burton for providing KZ52 expression plasmids.

# Author Contributions

**Conceptualization:** Dibyendu Kumar Das, James B. Munro.

**Data curation:** Dibyendu Kumar Das, Uriel Bulow, William E. Diehl, Natasha D. Durham, James B. Munro.

**Formal analysis:** Dibyendu Kumar Das, James B. Munro.

**Funding acquisition:** Jeremy Luban, James B. Munro.

**Investigation:** Dibyendu Kumar Das, Uriel Bulow, William E. Diehl, Natasha D. Durham, Fernando Senjobe, James B. Munro.

**Methodology:** James B. Munro.

**Resources:** Kartik Chandran.

**Supervision:** Jeremy Luban, James B. Munro.

**Writing – original draft:** Dibyendu Kumar Das, James B. Munro.

**Writing – review & editing:** Dibyendu Kumar Das, Jeremy Luban, James B. Munro.

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
