## [Editor Report · Decision Letter 0]

18 Sep 2019

Dear Dr Munro, 

Thank you for submitting your manuscript entitled "Conformational changes in the Ebola virus membrane fusion machine induced by pH, Ca2+, and receptor binding" for consideration as a Research Article by PLOS Biology.

Your manuscript has now been evaluated by the PLOS Biology editorial staff as well as by an academic editor with relevant expertise and I am writing to let you know that we would like to send your submission out for external peer review.

Please re-submit your manuscript within two working days, i.e. by Sep 20 2019 11:59PM.

Kind regards,

Lauren A Richardson, Ph.D

Senior Editor

PLOS Biology

---

## [Decision Letter · Decision Letter 1]

21 Oct 2019

Dear Dr Munro,

Thank you very much for submitting your manuscript "Conformational changes in the Ebola virus membrane fusion machine induced by pH, Ca2+, and receptor binding" for consideration as a Research Article at PLOS Biology. Your manuscript has been evaluated by the PLOS Biology editors, an Academic Editor with relevant expertise, and by several independent reviewers.

As you will read, the reviewers found your study generally well done and were intrigued by the results. However, they do raise some points that will need to be addressed in a revision. Of particular note, they request functional experiments testing the reversibility of GP conformational changes observed in smFRET, better controlling the DiO mixing assay, and an improved explanation of the effects of GP labeling on infectivity. Additionally, the Academic Editor suggests caution when interpreting the functional relevance of the observed smFRET states, given the inherent difficultly in linking smFRET and lipid mixing results.

In light of the reviews (below), we will not be able to accept the current version of the manuscript, but we would welcome resubmission of a much-revised version that takes into account the reviewers' comments. We cannot make any decision about publication until we have seen the revised manuscript and your response to the reviewers' comments. Your revised manuscript is also likely to be sent for further evaluation by the reviewers.

Your revisions should address the specific points made by each reviewer. Please submit a file detailing your responses to the editorial requests and a point-by-point response to all of the reviewers' comments that indicates the changes you have made to the manuscript. In addition to a clean copy of the manuscript, please upload a 'track-changes' version of your manuscript that specifies the edits made. This should be uploaded as a "Related" file type. You should also cite any additional relevant literature that has been published since the original submission and mention any additional citations in your response. 

Before you revise your manuscript, please review the following PLOS policy and formatting requirements checklist PDF: http://journals.plos.org/plosbiology/s/file?id=9411/plos-biology-formatting-checklist.pdf. It is helpful if you format your revision according to our requirements - should your paper subsequently be accepted, this will save time at the acceptance stage.

Please note that as a condition of publication PLOS' data policy (http://journals.plos.org/plosbiology/s/data-availability) requires that you make available all data used to draw the conclusions arrived at in your manuscript. If you have not already done so, you must include any data used in your manuscript either in appropriate repositories, within the body of the manuscript, or as supporting information (N.B. this includes any numerical values that were used to generate graphs, histograms etc.). For an example see here: http://www.plosbiology.org/article/info%3Adoi%2F10.1371%2Fjournal.pbio.1001908#s5.

For manuscripts submitted on or after 1st July 2019, we require the original, uncropped and minimally adjusted images supporting all blot and gel results reported in an article's figures or Supporting Information files. We will require these files before a manuscript can be accepted so please prepare them now, if you have not already uploaded them. Please carefully read our guidelines for how to prepare and upload this data: https://journals.plos.org/plosbiology/s/figures#loc-blot-and-gel-reporting-requirements.

Upon resubmission, the editors will assess your revision and if the editors and Academic Editor feel that the revised manuscript remains appropriate for the journal, we will send the manuscript for re-review. We aim to consult the same Academic Editor and reviewers for revised manuscripts but may consult others if needed.

We expect to receive your revised manuscript within two months. Please email us (plosbiology@plos.org) to discuss this if you have any questions or concerns, or would like to request an extension. At this stage, your manuscript remains formally under active consideration at our journal; please notify us by email if you do not wish to submit a revision and instead wish to pursue publication elsewhere, so that we may end consideration of the manuscript at PLOS Biology.

When you are ready to submit a revised version of your manuscript, please go to https://www.editorialmanager.com/pbiology/ and log in as an Author. Click the link labelled 'Submissions Needing Revision' where you will find your submission record. 

Sincerely,

Lauren A Richardson, Ph.D

Senior Editor

PLOS Biology

Reviews

Reviewer #1: 

Fusion stage of the entry of Ebola virus is a complex and still poorly understood process. Activation of viral fusogens in virus-carrying endosome depends on a number of factors, including cathepsin-mediated cleavage of envelope glycoprotein GP and Niemann-Pick C1 (NPC1) binding. It has also been suggested to depend on acidification and changes in the calcium concentration associated with maturation of endosomes. This study analyses conformational changes in fusogenic GP2 domain of GP using an elegant approach they have developed in their earlier work on flu hemagglutinin. The loss of initial conformation of GP2 is detected at a single molecule level by a single-molecule Förster resonance energy transfer (smFRET) imaging assay as a decrease in resonance energy transfer reflecting an increase in the distance between two fluorophores placed at positions that are close to each other in the initial conformation. Based on the smFRET data and a bulk assay that detects lipid mixing between DiO-labeled GP-pseudoviruses and liposomes as dequenching of DiO, the authors conclude that low pH and Ca2+ dependent reversible shifts of GP2 conformations to intermediate state(s) prime the protein for NPC1 binding dependent irreversible conformational change associated with fusion. While the results are mostly convincing, definitely novel and significant, and will be of importance for researchers working on entry stages of Ebola virus, and on fusion mediated by other protein fusogens, I do have a number of questions/suggestions. While the smFRET part of the work is very well designed, my questions are mostly related to a virus-liposome lipid mixing assay. I consider two of the major comments below as essential to support the current conclusions. 

Major comments. 

1) The DiO dequenching data are presented as the fold increase over initial fluorescence (F/F0). Liposome-based fluorescence dequenching fusion assays conventionally require application of a detergent at the end of the experiment to normalize the extent of the dequenching to the maximum dequenching with infinite dilution of the probe. Without it, we do not know the extent of DiO quenching in the viral particles prior to fusion and cannot evaluate the efficiency of the lipid mixing. Also, it appears that the extent of DiO dequenching is very very low (the largest increase is ~1.6 fold) comparing with what they had in the Cell paper for HA (Ref 35), where DiO fluorescence went from ~0 to 2000 in 150 s. Does it mean that GP2-mediated virus/liposome lipid mixing is much much less efficient than for flu? If this is the case, it is essential to verify that dequenching cannot be explained by fusion-unrelated lipid probe transfer. 

2) The dependence of the GP2 restructuring on Ca is one of the most intriguing features of the process. The effects of TPC inhibitors on Ebola virus infection and lipid mixing inhibition by 1 mM and higher concentrations of Ca on the lipid mixing suggests that fusogenic restructuring of GP2 depends on decrease rather than increase in Ca. In this scenario millimolar concentrations of Ca in the extracellular environment act as one of the blocks for premature restructuring and inactivation of the protein. Have you checked the reversibility of low pH and Ca effects in the context of virions and functional experiments as an inactivation? Do you observe an irreversible or reversible loss of fusogenic activity of the virus for higher concentrations of Ca (S2B Fig.), especially, prior to NPC1? 

Minor comments

1) I am confused by what appears to be a discrepancy between the main text and the legend to S1 Fig. The text states “Pseudovirions labeled with DiO maintained approximately 75% infectivity as compared to unlabeled pseudovirions” and the legend to the S1 Fig. states “Infectivity of GFP-expressing VSV particles pseudotyped with GP or GP*, or labeled with the lipophilic fluorophore DiD, which is spectrally distinct from GFP and chemically similar to the DiO fluorophore used in the hemifusion assay.”. Have the authors examined the infectivity for virions labeled with DiO (as in functional experiments) or with DiD?

2) I suggest the authors to refer to their DiO dequenching assay as lipid mixing assay rather than hemifusion assay. In the absence of other assays, we cannot distinguish hemifusion from complete fusion here because complete fusion would also result in lipid mixing and DiO dequenching. 

3) Line 411: “virus and liposomes were co-incubated at room temperature for 5 minutes to enable virus-liposome binding at pH 7.5.” What binding mechanism you have in mind?

4) I am unclear on whether the authors used 1-oleoyl-2-palmitoyl-snglycero-3-phosphocholine as stated in line 403 (Avanti # 850457) or POPC (line 403) (1-palmitoyl-2-oleoyl-glycero-3-phosphocholine) (Avanti #850475).

--------------

Reviewer #2: 

Das, D. K. et al present a thorough study of the role of pH, Ca2+, and target receptor binding for the Ebola virus glycoprotein spike. These authors use a smFRET approach to estimate the conformational dynamics of a domain in the spike under differing ligated conditions for the aforementioned species. The results demonstrate that these ions and the sNPC1 receptor fragment are required to yield a low FRET state, a conditions the authors speculate is the post-fusion state based upon atomic structures of the two states. This elegant and mechanistically insightful study should certainly be published after addressing this reviewer’s concerns.

Major points:

1) This reviewer envisions the sequence of events for GP encounter between the relevant players discussed in this study to occur in this order: Lower pH, lower Ca2+, additional pH reduction, and then a final encounter between the NPC1 luminal domain in a late endosome/lysosome (similar to the mechanistic model cartoon). The author’s experimental setup, however, appears to create a steady state with the addition of sNPC1 to GP (and variants) and then observe the change in smFRET relative to pH modulation or steady-state Ca2+ addition. Have the authors attempted to perform a time series smFRET measurement by perfusing in sNPC1 under acidic conditions and 0.5 mM Ca2+? This would help to rectify the proposed linear cartoon model, suggesting that low pH and Ca2+ are required a priori to initiate NPC1 receptor engagement. If this experiment were performed and the resulting occupancy distributions were nearly the same, this would suggest that the mechanism does not require a specific order of events, simply all of these ingredients are required.

2) The data in Figure 1 should be fit to a model and quantitative estimates should be extracted to report relative changes in rates of fusion compared with a standard condition (e.g. sNPC1 + Ca2+).

3) This reviewer wonders if the MD simulation of probes on GP can be used to extract the κ2 value for the dye pair orientation as a function of time to estimate a more accurate, albeit a simulated, explicit distance measurement. Does the distribution of dipolar angles resemble the often abused 2/3 random orientation value for κ2. It is hard to tell from the centroid dots how the dipoles are oriented in the cartoon. If the authors were to extract these values and estimate the mean κ2, would the FRET efficiency convert to distances that resemble those speculated in the paper?

Minor points:

1) For all figures displaying the fractional occupancy of a state via smFRET, it should be explained in the methods how the error in occupancy is calculated. Since this parameter is a fractional value and the fit to a multimodal distribution should have only one error, it is not clear how the error is propagated to these plots.

2) Quantitative color maps should be includes for all smFRET probability distributions. It is difficult to interpret the gradient changes in the time-lapse measurements.

3) Lines 125-126, has anyone measured the luminal concentrations of Ca2+ in late endosomes/lysosomes (the site of entry for EboV)? If so, what is the average molarity? Does this compare well with the 0.5 mM used in the study?

4) The authors should add a reference to line 152 to link the audience to what is known about the epitope of the KZ52 antibody.

--------------

Reviewer #3: 

The Das et al. manuscript studies Ebola GP fusion protein mediated membrane fusion using pseudovirons. They use a liposome fusion fluorescence dequenching method to reveal membrane hemifusion and single molecule FRET approach to reveal Ebola GP fusion protein conformational changes, as they previously did with HIV and influenza. The authors use the liposome assay to conclude that that low pH, receptor binding, and calcium exposure act in concert to induce the most efficient fusion activity, as is generally believed for Ebola from previous studies. They use the single molecule FRET method to reveal transitions from a prefusion configuration with high FRET to a low FRET state, which is considered to reflect an on-pathway fusogenic state. Correlations among hemifusion lipid mixing signals and single molecule FRET signals under various experimental conditions are used to support the interpretation of these FRET states as fusion-related conformations. The authors map the populations of events with the different FRET signals under several conditions including the removal of the glycan cap, receptor presence as well as pH and calcium exposure. Controls with different divalent ions, inactivating antibodies and drugs used to inhibit Ebola are included, strengthening the arguments. The most novel finding in the work is identification of reversibility in the low pH + calcium induced conformational change (transition to low FRET) that is made permanent in the presence of the cognate receptor. Finally, there is a general discussion of a model for Ebola membrane fusion in the context of existing models for influenza and HIV. 

The manuscript is clear and the experiments are expertly performed. The work represents a high level of technical sophistication. The topic of Ebola membrane fusion is important and interesting to a wide array of scientists. This approach to mechanistic studies of virus phenomena is exciting and likely to be enduring in the field. The model of several synergistic interactions regulating virus fusion and entry into cells is innovative. Nevertheless, there are a few concerns with the work that are essential to address.

1 It would be useful to confirm the observation of reversibility of the transition from prefusion configuration to fusogentic state (in the absence of receptor) in the liposome assay, if possible. Perhaps experiments with various sequences of pre-exposure to changes in pH/calcium with or without receptor present, and then assay for hemifusion competence in the liposome assay would be useful. Such experiments could be highlighted as supporting the existence of this reversible state that is on pathway supporting membrane fusion.

2 What is the labeling efficiency of the proteins in the single molecule FRET studies? Maybe this is hard to determine, but it is important for the infectivity and fusion validation work (Figure S1). Lines 151-155 describe measurements of labeled sample infectivity and fusion efficiency as ranging 60%-90% of unlabeled or wild-type. Labeling efficiency could correlate with these effects, suggesting the proteins lacking dyes could account for the active fraction of the population. 

3 These functionality controls (Figure S1) are also difficult to interpret given that lines 436-440 mention that the virus particles here had 50 fold excess of wild-type GP over the labeling mutant GP*. Does this mean each particle has more wild-type fusion proteins than labeled fusion proteins? Note line 158 mentions pseudovirions were formed with a single GP* protomer among the native distribution of wild-type GP, and line 341 in methods mentions full-length GP0* was translated at approximately 5-10% efficiency as compared to wild-type GP. These statements suggest this is the correct interpretation, which then questions the relevance of the controls – an excess of wild-type, unlabeled GP protein will mask any failure of the FRET reporting GP* in the event that the FRET modifications rendered the GP* or GP*+fluorophore protein not functional for fusion or infection. This information is important to the paper, but as presented is confusing and concerning.

4 The authors also observe a FRET event near 0.52 that is not claimed to be directly related to the pre-activation or fusogenic states. This state does not change under most conditions: line 170: “intermediate-FRET state (0.52 ± 0.09), which did not respond to changes in pH” or in figure 3 the intermediate-FRET state does not change with calcium either. That FRET 0.52 state is similar in all conditions and all mutants in figure 3 (20-30% populations). The authors only comment once to address possible ways to interpret it, saying near line 254 – “The intermediate-FRET state may reflect a more localized rearrangement of the fusion loop that precedes its large-scale displacement”. Is it reasonable that this FRET 0.52 state is related to anything functional if this state does not change with pH or calcium? The authors should try to understand this state more completely.

--------------

Reviewer #4: 

The authors have created a construct of the Ebola GP protein that allows Cy3 to be attached to one site, and Cy5 to another. If there are changes in the distance between these two dyes, their dipolar orientation, the dielectric constant of the medium between them, or other changes to the electronic configuration of either dye, there will be a change in the FRET signal. They measure the both donor and acceptor fluorescence from single molecules as a function of time after changing pH (before they bleach) when the proteins are expressed on pseudoviruses and added to liposomes bearing receptor. When the pH is changed, as modulated by Ca2+, proteolytic cleavage, or addition of soluble receptor, the average FRET can change. These changes are categorized into three states for which gaussian distributions are reported, normalized to these three states. In addition, lipid dequenching is measured as a function of time with these same viron under conditions varied as above, as a measure of fusion or hemifusion.

There is much good that this paper contains, with a lot of very good and careful work. I cannot say the same about the writing or even the final conclusion. Instead of leaving the interpretation of data for the Discussion, they intersperse it into the Results, making a very complicated first paragraph with not much interpretation of results, or a summary of the main experimental findings. They can certainly propose a model, but first they need to discuss their results. They would be well advised to move all interpretation into the Discussion.

End of Abstract and lines 270-276: It is not clear that the data here suggests protection of internalized Ebola in early entry until a “more permissive environment” is reached – presumably maturing of early endosomes. The one paper I found (Albrecht et al. Cell Calcium 57:263-274, 2015) with good endosomal targeting and genetic-encoded sensor detection of endosomal Ca++ suggests the opposite, since endosomal Ca++ starts below 5 µM and falls even lower with maturation. Thus lines 270 – 276 seem speculative as there are no experiments in this paper showing the actual Ca++ in the endosome is actually protective, and is represented within the ranges of the entire biphasic response to Ca++. The cited (Fan et al.) paper on endosomal calcium and SERM and Ebola indicates that chelating Ca blocks fusion, but even the SERM which also block Ebola only double Ca++ release, but they have to somehow bring endosomal Ca++ from 2 µM to over 1 mM??? So the biphasic nature of the Ca response is not sufficiently fleshed out in this paper to end the first paragraph of the Discussion with this statement. Furthermore, it seems that cholesterol is a better candidate for SERM action.

Lines 300-302. The last sentence of the paper is very weak. These are not novel ideas and come across as obvious given the involvement of many receptor/fusogen binding reactions that potentiate conformational changes downstream of binding. Surely with such great stuff they can do better. They make it clear in the introduction that far too little is known, and after reading this manuscript I am sure that their hypothesis in line 180-182 is very likely a reasonable and testable hypothesis. This is a solid step forward: “According to this

180 interpretation, acidification promotes a conformational change in GP, which releases the

181 fusion loop from the hydrophobic cleft in the neighboring protomer, perhaps facilitating

182 its interaction with the target membrane.” It can be an important concluding model, as fleshed out with their reversibility data vis-à-vis the other intermediates. 

Line 246. Holistic is going a bit too far for my taste. It is not clear what is the effect of other factors. A new assay is devised for protein conformational change and lipid mixing is measured in permutations of Ca, pH, for two constructs with and without soluble receptor. Not exactly all the interacting factors are integrated into one model. Nor is it particularly surprising that pH causes a conformational change in a viral fusion protein. Moreover, terms appear like ‘physiological’ which indicate native structures, but in this report the work is done with engineered constructs, usually derivatized both genetically and chemically, and synthetic liposomes.

Other issues:

1. DiO dequenching is a lipid-mixing assay that occurs in both hemifusion and fusion. For determination of hemifusion or fusion a content mixing assay is also required. Also, lipid dequenching assays for lipid mixing are usually followed by a detergent step to determine maximal dequenching. Then the % dequenching is plotted. I think that gives a better sense of the efficiency or robustness. Here it is relative, so it is not clear what fraction of the viron give lipid mixing at 1.6 F/F0. I could be 1 % or 100%.

2. Lines 129-131- “…cannot rule out…does not proceed...” Double negative, does it mean ‘we rule out that it does proceed’? I think this indicates the above problem: hemifusion is the merger of membranes without fusion pore formation, so what is measured here is lipid mixing. It is an aspect of both hemifusion and fusion. It should be called what it is, and it should not be called either hemifusion or fusion, as neither is shown at this point. Usually, the interpretation of the result is left for the discussion section. Changing the standard definition of hemifusion leads to awkward sentences like this one. By calling the observable ‘lipid mixing’, the authors can use an understandable sentence like:

"At present, we cannot determine what fraction of the lipid mixing corresponds to full fusion or hemifusion."

3. Line 476. There are at least 3 SPARTAN software packages on the web, but the first 40 or so references to SPARTAN software are to another package. I see the correct citation as requested but I do not see on online access to the custom-written scripts. As this is not open sourced but licensed I suggest that the url be provided in parentheses after the SPARTAN (https://www.scottcblanchardlab.com/software) and the custom written scripts be available to readers.

4. Line 752 - There is no such journal ‘Sci Rep-uk’. The citation is ‘Scientific Reports, 7:41226’ However PLOS wants to abbreviate or format it.

5. Figure 1. ‘Efficiencies are different’. It is not clear, since the red signal continues to grow, if lipid mixing efficiency is actually different, one would have to wait. Clearly the rate of lipid mixing is different, but that is what you get when you compare two time points, not the extent of lipid mixing (I assume you mean extent by 'efficiency'.

6. Figure 3, 4, etc. There is something odd about the plots. By definition, FRET goes from 0 to 1. Yet this FRET goes above 1. How can that be?

7. Figure 4. Maybe the phrasing "not predominantly high..." means "not all the FRET came back to the high FRET state". You probably want to state in the figure legend that ‘time’ in the figure (in seconds) refers to the time after the stop flow step and not the incubation times.

8. Fig. 5: I have a hard time understanding this cartoon until I saw it was a model. I thought the authors were summarizing their results (not a bad idea, because there are lots of great results). The experiments first mixed the liposomes and viruses and Ca, and only later added low pH. The npc1 was added to the liposome virus mixture. So all the data just what happens in seconds after adding acid, whereas the cartoon in Fig. 5 shows H+ and Ca++ added before the NPC1.

9. Line 406. What cation in the phosphate buffer for making liposomes?

10. Line 409. What excess of soluble receptor? Was it washed off before liposomes were used? How do we know it was the receptor on the liposome that made for lipid mixing and not the excess?

---

## [Editor Report · Decision Letter 2]

30 Dec 2019

Dear Dr Munro,

Thank you for submitting your revised Research Article entitled "Conformational changes in the Ebola virus membrane fusion machine induced by pH, Ca2+, and receptor binding" for publication in PLOS Biology. 

The Academic Editor and I have now assessed your revision and we're delighted to let you know that we're satisfied with your manuscript. The Academic Editor does note one concern, which is that in Figure 2A, which is supposed to depict a pseudovirus with GP glycoproteins protruding from the viral membrane, the viral membrane is not visible and GPs appear to float in the space. We will very likely publish your study, assuming you are willing to address this concerns and make the final edits needed to meet our production and data requirements.

Before we can formally accept your paper and consider it "in press", we also need to ensure that your article conforms to our guidelines. A member of our team will be in touch shortly with a set of requests. As we can't proceed until these requirements are met, your swift response will help prevent delays to publication. Please also make sure to address the data and other policy-related requests noted at the end of this email.

*Copyediting*

*Published Peer Review History*

*Early Version*

*Submitting Your Revision*

Sincerely,

Lauren A Richardson, Ph.D

Senior Editor

PLOS Biology

DATA POLICY:

2) Deposition in a publicly available repository. Please also provide the accession code or a reviewer link so that we may view your data before publication. **Please deposit all structures and list accession codes in the Data Statement.**

1, 3A-C, 4A-C, S1AB, S2A-D, S3, S6, S7, S8, S9B

---

## [Editor Report · Decision Letter 3]

23 Jan 2020

Dear Dr. Munro,

On behalf of my colleagues and the Academic Editor, Dr. Gregory Melikyan, I am pleased to inform you that we will be delighted to publish your Research Article in PLOS Biology. 

Early Version

PRESS 

Kind regards,

Krystal Farmer,

Development Editor 

PLOS Biology

on behalf of

Lauren Richardson,

Senior Editor

PLOS Biology